# Current trends in AI tools development in plant biology
# Conference Submissions

## Abstract

AI tools have revolutionized plant biology and agrobiology by enabling high-throughput analysis of complex biological data, from genomes to phenotypes, supporting precision breeding and sustainable agriculture. We highlight key applications in bioimage analysis using YOLO algorithms, crop plant genome analysis, and computational high-throughput phenotyping tools, discussing recent advancements. Classical genetic selection demands long time and labor-intensive opening way to novel computational modeling. Modern omics technologies, high-throughput genomics, high-throughput plant phenotyping, development of remote sensing devices on-farm provided vast amount of data available in databases. Such platforms and associated data generation have contributed to a booming AI industry in agriculture. Computer methods for plant genome data analysis oriented on stress resistance and crop yield rely on AI approaches including pattern recognition, neural networks, and LLMs. As an example of the development of AI tools for crop bioinformatics, we present a collaborative project between Russia and China "Smart Crop - Cognitive Platform for Reconstruction, Visualization and Analysis of Stress Response Networks based on ANDSystem and Multiomics in Rice and Wheat". The study of molecular genetic mechanisms of plant resistance to unfavorable biotic and abiotic factors (high or low temperature, drought, salinity, soil diseases, pathogens and pests) requires the study of the functioning of entire molecular genetic systems, including complex signaling, regulatory, transport and metabolic pathways. Finally, we discuss the challenges and current trends of AI applications in computational plant biology.

## 1 Introduction. Plant Genome Analysis and Project 'Smart Crop'

Artificial intelligence (AI) is revolutionizing plant sciences by enabling accurate identification of plant species, early disease diagnosis, prediction of crop yield, and precision agriculture optimization (Gupta et al., 2024; Khan et al., 2025; Kassem, 2025). AI applications in plant research and agriculture have benefited large-scale industrial farming, with RD investment focused on commodity crops such as wheat, rice, and maize, and high-value horticulture crops such as soft fruits. The challenge is related to its importance for global food security. Extreme environmental impacts, including those caused by adverse climate changes, plant diseases and insect pests, lead to significant crop losses. AI-driven multi-omics integrates genomics, transcriptomics, and metabolomics for crop improvement, using deep learning to predict traits like yield and stress resistance (Javaid et al., 2023). Models such as convolutional neural networks analyze pan-genomes in crops like wheat and maize, identifying QTLs (Quantitative Trait Loci) for seed quality and accelerating breeding. Recent reviews highlight AI's role in plant genomics, where ML dissects regulatory networks and enhances genomic selection for climate-resilient varieties (Eftekhari et al., 2024; Khan et al., 2025).

Within the framework of the project, using artificial intelligence methods, a cognitive software and information platform "Smart Crop" was created, which provides a full cycle of knowledge engineering in the field of plant biology for the reconstruction and analysis of gene networks, focused on solving problems of studying the molecular genetic mechanisms of the genotype-phenotype relationship - medium for agriculturally valuable crops of rice and wheat (Cheng et al., 2024). The platform

is focused on solving such substantive tasks as interpreting the results of omics data (establishing links between gene sets and biological processes, phenotypic traits, etc.); reconstruction of gene networks describing the relationship between molecular genetic objects and objects corresponding to the concepts of selection, phenomics and seed production, phytopathology, diagnostics, means of protection; identification of regulatory and signaling pathways of plant response to specific growth conditions, biotic and abiotic stresses (high or low temperature, drought, salinity, soil contamination with metals, response to fertilizers, the action of hormones, etc.); prediction of candidate genes for genotyping; search for markers for marker-based selection; candidate targets for the directed action of substances (including external factors) on plants to solve the problems of early/uniform emergence, better vegetative growth, efficient absorption of nutrients (Kleshchev et al., 2024; Antropova et al., 2024).

## 2   CROP PLANT GENOME ANALYSIS

Extreme environmental effects, diseases and insect pests result in significant crop loss. Therefore, more researchers turn their attention to the study of crop resistance. Earlier present, research on single stress-responsive genes in crops has been performed. To date, multiple studies have systematically revealed a part of molecular regulatory network in rice and wheat (Li et al., 2025). Recently, Artificial intelligence (AI) has opened new possibilities for the systematic analysis of molecular networks. Text-mining is a field in AI that aims to extract information from collections of text documents based on machine learning and natural language processing (NLP) techniques. Text mining is considered a useful tool for integrative biological research involving genes, proteins and phenotypes. In this project we plan to apply ANDSystem - bioinformatics tool that builds molecular networks by text-mining and data-mining of PubMed indexed publications and multiple omics by Russian partners (Ivanisenko, et al., 2020; Demenkov, et al., 2011; Ivanisenko, et al., 2022), for reconstruction and analysis of molecular networks under specific growth conditions, biotic and abiotic stresses in rice and wheat.

Therefore, the "Smart Crop" cognitive software and information platform was created, providing a full cycle of knowledge engineering about the genotype-phenotype-environment relationships in the field of agriculturally valuable rice and wheat crop research using AI and text-mining methods. The platform included three modules: (1) a module for automated extraction of knowledge from the texts of scientific publications, patents, databases; (2) a knowledge base containing extracted information in the form of a knowledge graph; (3) a module designed for the reconstruction and analysis of gene networks that describe the molecular mechanisms, regulatory and signaling pathways of a plant's response to specific growth conditions, biotic and abiotic stresses (Demenkov et al., 2025).

Non-coding RNA (ncRNA) study is an important research field in life sciences. ncRNAs are involved in various biological processes and molecular regulations. Based on AI applications on plant ncRNAs and their regulatory network construction, we explored high-throughput sequencing datasets such as Ribo-seq, RNA-seq, sRNA-seq and other genomic/epigenomic data, to develop and improve molecular networks under stress response in rice and wheat (Shen et al., 2024; Zhou et al., 2025).

## 3   BIOIMAGE ANALYSIS WITH YOLO (YOU ONLY LOOK ONCE) MODELS

YOLO (You Only Look Once) models, known for real-time object detection, have been adapted for plant bioimaging to identify diseases, pests, and structures with high speed and accuracy (Alhwaiti et al., 2025). YOLOv3 and YOLOv4 excel in localizing diseased areas on leaves, achieving superior mAP, precision, and recall compared to traditional methods, aiding early intervention in crops like fruits. Recent 2025 studies demonstrate YOLO's integration with multispectral imaging for non-invasive trait extraction, enhancing biodiversity monitoring and precision agriculture.

New approaches for detecting plant diseases, based on image processing, Machine Learning (ML), Deep Learning (DL) techniques, and hyperspectral imaging have been developed as a result of technological advancements (Kaur et al., 2025). For instance, the applications included multi-class SVM for detecting three distinct classes of apple diseases, a feed-forward back propagation neural network for detecting mildew from grape leaves, machine learning techniques for the prediction

Table 1: Recent works in AI – driven plant bioinformatics (2024-2026)

| Area | Key contribution | Publication |
|---|---|---|
| Bioimage | Real-time detection | Alhwaiti et al. 2025 |
| Genome | Deep networks for omics | Syeda, 2025 |
| Inter-disciplinary | GPT model for plants | Zhang et al., 2025 |
| Phenotyping | Synthetic data for trait algorithms | Roggiolani et al., 2025 |
| Genome/ Phenotype | Multi-omics for breeding values | Kassem, 2025 |
| Phenotype | Phenotyping in field using drones | Lu et al., 2025 |
| Inter-disciplinary | Plant disease diagnosis | Sheikh et al., 2024 |
| Phenotyping | Yield prediction | Pugh et al., 2024 |

of crop yield. DL-based approaches for detecting plant diseases have attracted growing interest in recent years due to their capacity to acquire complex representations from vast amounts of data.

YOLO-LeafNet approach is proposed for detecting diseases from leaf images of four distinct species, namely, grape, bell pepper, corn, and potato. About 9 thousand leaf images have been acquired for this work from five different publicly available datasets on Kaggle. All the acquired images were pre-processed by applying four different image pre-processing operations. The number of images in the training dataset was tripled for better model performance by applying five different augmentation operations.

# 4 PLANT PHENOTYPE ESTIMATION TOOLS

Computer vision tools automate phenotyping, quantifying growth, morphology, and physiology from images and 3D models (Kassem, 2025). Recent generative AI creates synthetic 3D leaf point clouds for trait estimation, addressing data scarcity and improving yield predictions (Plant Phenomics, 2025). Platforms like those using CNNs enable field-scale analysis via drones, modeling genotype-phenotype links for nowcasting and forecasting. Tools such as scKAN and AI phenomics pipelines integrate multi-omics for precise seed trait mapping

The study by Roggiolani and colleagues (Roggiolani et al., 2025) introduces a generative model capable of producing lifelike 3D leaf point clouds with known geometric traits, accelerating crop improvement and optimize yield predictions through data-driven modeling.

The research team trained a 3D convolutional neural network to learn how to generate realistic leaf structures from skeletonized representations of real leaves. Using datasets from sugar beet, maize, and tomato plants, they extracted the "skeleton" of each leaf—the petiole and main and lateral axes that define its shape—and then expanded these skeletons into dense point clouds using a Gaussian mixture model. The neural network, designed as a 3D U-Net architecture, predicts per-point offsets to reconstruct the complete leaf shape while maintaining its structural traits.

AI in plant breeding faces significant hurdles like data scarcity and model interpretability but holds promise for predictive breeding and climate resilience. Recent reviews emphasize addressing these to unlock faster genetic gains (Plant Phenomics, 2025).

# 5 FIELD SURVEILLANCE USING DRONES

Digital phenotyping using drones (Unmanned Aerial Vehicles - UAVs) enables high-throughput, non-destructive trait assessment in breeding plots, accelerating selection for yield, stress tolerance, and maturity (Eftekhari et al., 2024). These applications integrate multispectral/RGB imaging with AI for scalable field data collection. Drones quantify key traits like plant height, biomass, stand count, canopy cover, and relative maturity with 90 percent accuracy, replacing manual labor in crops such as maize, soybean, and dry beans. RGB and multispectral sensors capture vegetation indices (e.g., NDVI) at multiple growth stages, strongly correlated with agronomic performance, such as yield and height.

UAVs detect pests, plant diseases and abiotic stresses via hyperspectral imaging, enabling early selection of resistant genotypes in breeding nurseries. High-resolution imagery identifies symptoms at the plot level, supporting genomic selection for traits such as drought tolerance through digital twins.

For example, in soybean breeding, drone HTP refines genotype selection via vegetation indices, optimizing research cycles and reducing costs. Cost-effective solutions like VITO's allow in-house deployment for large-scale trials, enhancing precision in variety development. Drones bridge genomics-phenomics gaps, simulating analysis of genotype-environment interactions for predictive breeding.

# 6 KEY CHALLENGES OF AI IN PLANT BIOLOGY

Data management is a primary barrier, with issues in standardization, quality, and availability of high-throughput phenomics and multi-omics datasets, especially for orphan crops (Williamson et al., 2023). High computational demands, biases in training data, and lack of field validation hinder model reliability and scalability. Interpretability remains elusive, as "black-box" AI struggles with explaining predictions for regulatory approval, while ethical concerns like inequality in access and GMO risks persist.

Note current challenges in AI applications in agrobiology:

Data scarcity/quality

Federated learning, synthetic data generation

Computational cost

Field-to-lab

Hybrid AI-process-based models will integrate mechanistic simulations with ML for better extrapolation to new environments and traits. Advances in high-throughput phenotyping via drones, robotics, and digital twins promise real-time, scalable trait assessment. AI-driven gene editing (e.g., CRISPR optimization) and predictive simulations could shorten breeding cycles to months, enabling proactive designs for stress tolerance.

ACKNOWLEDGMENTS

The work was supported by Russian Science Foundation and NSFC.

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
