# OpenReview forum: "Current trends in AI tools development in plant biology"
_mathai.club/MathAI/2026/Conference — MathAI 2026 Conference Submission_

### Official Review · Reviewer_gE1H · 2026-03-12
**Small number of technical details**

**Rating:** 6
**Confidence:** 3

**Review:**

The article "Current trends in AI tools development in plant biology" discusses the latest advances and applications of AI in plant biology.

The article's strength is its comprehensiveness. It covers numerous problems in the subject area, provides references to relevant work on these problems, and describes the key challenges of implementing neural networks in this field. An example of a proprietary development, the "Smart Crop" system, is also provided.

In my opinion, the article lacks technical details, information on advanced architectures, datasets, training methods, and comparisons based on metrics, speed, and resource consumption.

One notable criticism regarding the formatting is the lack of a link in the text to Table 1, which presents the latest work in AI.

---

### Official Review · Reviewer_6hBc · 2026-03-12
**Current trends in AI tools development in plant biology conference submissions**

**Rating:** 3
**Confidence:** 4

**Review:**

This manuscript surveys recent developments in AI tools applied to plant biology, with emphasis on applications in crop improvement, disease detection, phenotyping, and genomic analysis. The authors discuss the use of YOLO models for plant disease detection from leaf images, deep learning for multi-omics data integration, drone-based phenotyping, and the "Smart Crop" platform that combines text mining and knowledge graphs to study stress response networks in rice and wheat. The paper includes a table summarizing recent works (2024–2026) and a list of references (25 items).

### Major Concerns

1. **Mathematical Rigor (Score: 2)**
   The paper contains no mathematical formalism, theorems, proofs, or algorithmic details. It merely mentions that YOLO, CNNs, and other AI methods are used, without any explanation of their mathematical foundations. There is no discussion of loss functions, optimization, convergence, or statistical guarantees. The content is entirely descriptive.

2. **Novelty & Contribution (Score: 3)**
   The paper is essentially a literature review that compiles recent publications in the field. The only potentially novel element is the brief description of the "Smart Crop" project, but this is presented at a high level without technical depth. No new methods, datasets, or insights are introduced. The paper does not advance the state of the art.

3. **Relevance to MathAI (Score: 4)**
   While the paper deals with AI applications, it does not engage with the mathematical aspects of AI. The conference focuses on the mathematics of artificial intelligence—topics such as optimization theory, statistical learning, probabilistic modeling, and theoretical foundations. This paper is purely application-oriented and lacks any mathematical contribution, making it a poor fit for the conference's scope.

4. **Technical Quality (Score: 3)**
   The paper is a high-level survey with no experimental results, code, or data. The descriptions of methods are superficial (e.g., "YOLO models are used for detection"). The "Smart Crop" project is mentioned but not evaluated. There are no comparisons, benchmarks, or quantitative assessments. The references are numerous but the paper does not critically analyze or synthesize them.

5. **Clarity & Presentation (Score: 6)**
   The paper is reasonably well-organized and readable, though some sections are repetitive (e.g., the abstract repeats itself). The table of recent works is helpful. However, the language is occasionally awkward, and there are minor formatting issues. Overall, the exposition is adequate for a review article.

6. **AI-Generation Risk (Score: 3)**
   The paper appears to be human-written. It contains specific references to a collaborative project and mentions authors' own work (e.g., Ivanisenko et al.). The style is consistent with conference proceedings. There are no obvious signs of AI generation.

### Pros
- Timely and relevant topic: AI applications in plant biology are important for food security.
- Provides a broad overview of recent literature, which could be useful for newcomers to the field.
- Mentions a real-world collaborative project ("Smart Crop") that may be of interest.

### Cons
- Lacks mathematical content—no formulas, theorems, or algorithmic analysis.
- No original research contribution; it is a survey/review.
- No experimental validation or technical depth.
- Poor fit for a conference on the mathematics of AI.
- Superficial treatment of each topic; reads like an extended abstract.

### Recommendation
The paper does not meet the standards of MathAI 2026. It is a descriptive review with no mathematical substance. It would be more suitable for a biology or agriculture conference with a focus on applications, or as a short abstract in a workshop. For this conference, the paper should be rejected.

---

### Official Review · Reviewer_9yc2 · 2026-03-12
**Absence of mathematical substance and research contribution**

**Rating:** 6
**Confidence:** 4

**Review:**

This paper is a descriptive literature survey on AI applications in plant biology (disease detection, phenotyping, multi-omics, stress networks), including a brief mention of the "Smart Crop" platform. It lacks any mathematical formalism, theorems, proofs, algorithmic details, loss functions, or theoretical analysis, making it a poor fit for a conference focused on the mathematics of AI. The work contains no original methods, experiments, benchmarks, or technical depth it is essentially a high-level review of recent publications. While the topic is timely and the presentation is reasonably clear, the absence of mathematical substance and research contribution requires additional work on this topic.

---

### Official Review · Reviewer_Wm4h · 2026-03-12
**Interesting Project Proposal, but Lacks Empirical Validation and Cohesion**

**Rating:** 4
**Confidence:** 4

**Review:**

This paper presents a broad interdisciplinary review of artificial intelligence applications in plant biology, highlighting how computer vision, multi-omics, and UAVs can accelerate precision breeding. The authors introduce "Smart Crop," a cognitive software platform that utilizes NLP and text-mining for reconstructing stress response gene networks in crops like rice and wheat. However, the manuscript functions primarily as a descriptive project proposal lacking original empirical experiments, quantitative evaluations, or in-depth technical analysis of the discussed models. Most critically, the authors severely violate the double-blind review policy by explicitly detailing a collaborative project between Russia and China and acknowledging specific funding sources. Consequently, while the conceptual overview of AI challenges in agrobiology is relevant, the absence of empirical validation and the severe formatting violations fundamentally undermine the submission.
Strengths: Addresses the highly relevant intersection of AI, climate-resilient agriculture, and precision breeding.​
Provides a comprehensive, multi-level overview spanning from text-mined gene networks to UAV-based field phenotyping.​
Introduces a practical framework ("Smart Crop") that integrates genomics, transcriptomics, and metabolomics.​
Clearly identifies valid systemic challenges in the field, such as data scarcity and the "black-box" nature of AI models.​

Weaknesses: Critical violation of the double-blind review policy by revealing geographic affiliations and specific grant foundations.
Mostly conceptual with a complete lack of original experimental validation, performance metrics, or comparative analysis.​
Fragmented structure that reads more like a grant report than a cohesive scientific paper.​
Inconsistent citation formatting between the main text and the numbered reference section.​